# Connecting Images through Sources: Exploring Low-Data, Heterogeneous Instance Retrieval

**Dimitri Gominski** [1,2,*]**, Valérie Gouet-Brunet** [1] **and Liming Chen** [2]

[1] LaSTIG, IGN-ENSG, Gustave Eiffel University, 77420 Champs-sur-Marne, France; valerie.gouet@ign.fr
[2] LIRIS, École Centrale de Lyon, 69134 Écully, France; liming.chen@ec-lyon.fr
[*] Correspondence: dimitri.gominski@ign.fr

**Abstract:** Along with a new volume of images containing valuable information about our past, the digitization of historical territorial imagery has brought the challenge of understanding and interconnecting collections with unique or rare representation characteristics, and sparse metadata. Content-based image retrieval offers a promising solution in this context, by building links in the data without relying on human supervision. However, while the latest propositions in deep learning have shown impressive results in applications linked to feature learning, they often rely on the hypothesis that there exists a training dataset matching the use case. Increasing generalization and robustness to variations remains an open challenge, poorly understood in the context of real-world applications. Introducing the ALEGORIA benchmark, containing multi-date vertical and oblique aerial digitized photography mixed with more modern street-level pictures, we formulate the problem of low-data, heterogeneous image retrieval, and propose associated evaluation setups and measures. We propose a review of ideas and methods to tackle this problem, extensively compare state-of-the-art descriptors and propose a new multi-descriptor diffusion method to exploit their comparative strengths. Our experiments highlight the benefits of combining descriptors and the compromise between absolute and cross-domain performance.

**Keywords:** CBIR; cross-domain; cultural heritage; benchmarking; diffusion

## 1. Introduction

There exists, in particular in Europe, many datasets of aerial and terrestrial images describing the territories at different time periods, available in GLAMs (Galleries, Libraries, Archives, and Museums). They represent a unique heritage able to describe cultural and natural landscapes, landmarks, and their evolution with interesting viewpoints ("bird eye" for oblique imagery, "close environment" for terrestrial imagery) which may be very powerful for research in SSH (Social Sciences and Humanities) or for the industry. Those collections show many common geographical areas (as in the collections considered in the ALEGORIA benchmark), while they are usually organized in silos within GLAM institutions and then poorly interconnected and lacking of global structure.

More generally, in recent years, the massive deployment of digitization technologies and the increasing availability of digital data describing the past have made of those "big data of the past" a major challenge for research in information science and digital humanities. This observation can be illustrated by the growing popularity of national and international initiatives, such as E-RIHS (European Research Infrastructure for Heritage Science, erihs.fr (accessed on 1 July 2021)) or the LSRI (Large Scale Research Initiative) Time Machine (timemachine.eu (accessed on 1 July 2021)) that gather hundreds of institutions in Europe.

In this work, we approach the problem of interconnecting cultural heritage image data from a purely 2D, content-based point-of-view. This image-based approach can serve as an entry point before engaging further towards complex modelization: 3D models rely

on image localization and pose estimation [1]; 4D models (including time) need multiple views through time [2]; 5D models (time and scale) additionally make use of varying level of details available through various sources to build advanced representations [3]. Gathering and interconnecting image data are an essential starting step towards a better understanding of our cultural heritage, be it through dating content by reasoning [4], following the evolution of an area [5], reconstructing lost monuments [6], or visualization in a spatialized environment [7].

The content-based image retrieval (CBIR) community has designed a variety of tools, powered by recent advances in computer vision and deep learning. However, deep learning being by current design heavily dependent on training data, current technical propositions answer to the problem under the assumptions that 1. a large training dataset is available, 2. the testing dataset has semantics close to the training dataset, and 3. the data has low representation variability.

Cultural, historical, and geographical data constitute a case where it might occur that none of these assumptions is met. Grouped in collections hosted in institutions, images from different data sources are often heavily skewed in terms of class distributions, characterized by specific representation conditions (linked to the capturing technology) and lacking annotations, making it all the more important to connect them to properly exploit them. The key properties that we expect from feature extraction models here are invariance to representation variations and generalization ability, while maintaining high accuracy.

In this work, we introduce the problem of **low data, heterogeneous instance retrieval**, and show how it is a perfect example of a major obstacle for computer vision applications: the domain gap. Our contributions are as follows:

- We present the ALEGORIA dataset, a new benchmark made available to the community highlighting a panel of variations commonly found in cultural data through collections of vertical aerial imagery, oblique imagery, and street-level imagery through various sources.
- We propose new indicators for measuring cross-domain performance in image retrieval.
- We review available methods in the literature with the goal of identifying promising methods to solve our challenge. We reimplement and train two of these methods.
- We evaluate a panel of state-of-the-art methods on the ALEGORIA benchmark through three evaluation setups, and further compare their performance against known variations.
- Motivated by uneven performance of descriptors depending on image characteristics, we propose a new multi-descriptor diffusion method with a variation allowing fine-tuning of inter- and intra-domain performance.

This paper extends our previous work presented in [8], where we compared a set of state-of-the-art descriptors against classes with predominant variations for contents dedicated to aerial iconographic contents. We concluded that deep descriptors do offer unprecedented margins of improvement, notably thanks to attention and pooling mechanisms, but are very dependent on the training dataset. We highlighted the fact that heterogeneity remains an open problem, calling for datasets, benchmarks, and methods to propose solutions. The ALEGORIA dataset already proposed in [8] has been updated and enhanced with new annotations and evaluation protocols, and is fully presented and available to download (for research purposes only) at the address: alegoria.ign.fr/benchmarks (accessed on 1 July 2021). In this work, we go further with a broader review of methods handling the domain gaps and a new diffusion strategy for linking such methods.

This paper is organized as follows: in Section 2, we present the ALEGORIA benchmark, review the literature along the supervision axis, present the compared methods, and our proposed diffusion mechanism. In Section 3, we conduct various experiments that are discussed in Section 4 before the conclusions.

## 2. Materials and Methods

In this section, we present all the materials and methods used to support our analysis of cross-domain content-based image retrieval. Section 2.1 presents the ALEGORIA benchmark, some statistics and examples, and the evaluation protocol with new measures that we propose for cross-domain evaluation. Section 2.2 establishes important steps and notions for content-based image retrieval, with a framework serving as context for our literature review. In Section 2.3, we present key ideas and promising paths regarding the lack of specific training data, the first characteristic problem in ALEGORIA. In Section 2.4, we continue our review with ideas regarding the cross-domain problem, the second characteristic problem in ALEGORIA. Section 2.5 concludes by presenting the state-of-the-art methods we compare and our technical contributions.

### 2.1. ALEGORIA *Benchmark*

In this section, we present the ALEGORIA benchmark: Section 2.1.1 is dedicated to the description of the image data and their attributes, while Section 2.1.3 details the evaluation protocol associated with this dataset. Figure 1 shows examples drawn from the benchmark.

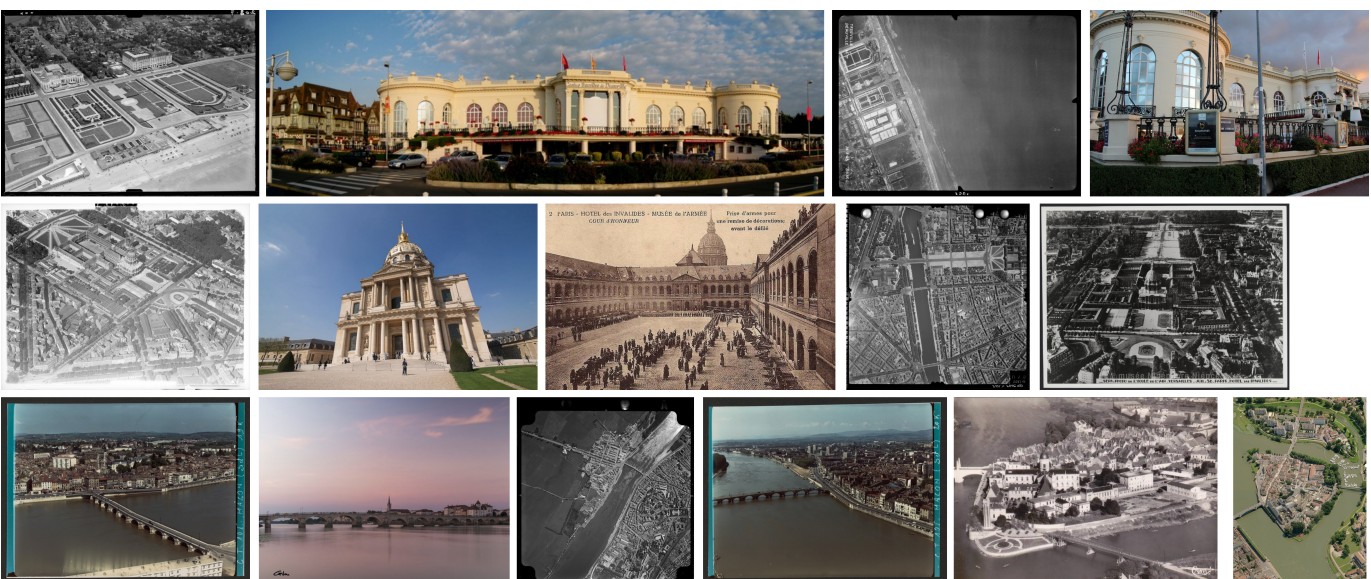

**Figure 1.** Samples from the ALEGORIA benchmark. Images in the same row belong to the same class.

#### 2.1.1. Presentation

The ALEGORIA dataset contains cultural and geographical images of various objects of interest in urban and natural scenes through a period ranging from the 1920s to nowadays. In collaboration with three GLAM institutions in the process of digitizing their content, the French Mapping Agency (IGN), the French National Archives (AN), and the Nicephore Niepce museum (NN), the images come from five image collections with different acquisition conditions:

- Henrard (NN): Oblique aerial imagery in the Paris region between 1930 and 1970,
- Lapie (AN and NN): Oblique (low and medium altitude) aerial imagery between 1955 and 1965,
- Combier (NN): Ground-level and oblique aerial imagery between 1949 and 1974,
- MRU (AN): Oblique aerial wide area imagery between 1948 and 1970,
- Photothèque (IGN): Oblique and vertical imagery between 1920 and 2010.

These collections all have the common characteristic of describing the French territory, but have sparse and non-standardized annotations that make them difficult to exploit "as-is". We added images downloaded from the internet (with permissive rights), i.e., mostly ground-level photography from the 2000s to today, to add variety in the representation

characteristics, provide useful intra-class "anchor" images (images that unambiguously depict the object or location of interest) and more generally insert links between cultural heritage data and the currently dominant volume of images, personal photography taken with smartphones or cameras.

The benchmark contains 58 classes defined with a geolocation (making it also suitable for image-based geolocalization), independently of how the image was captured (or drawn), with varying times of acquisition, vertical orientation, scales, etc. We refer to these influencing factors as variations, and propose to measure their influence by annotating each image with a set of (representation) attributes characterizing how the objects are represented. The dataset consists of 13,174 images, of which 1858 are annotated both with class labels and representation attributes. Table 1 presents general statistics about the dataset. For more extensive statistics, please refer to Table A1 (Appendix A) for how classes are defined or to the README available at alegoria-project.ign.fr (accessed on 1 July 2021).

**Table 1.** ALEGORIA benchmark statistics.

| Item | Value |
|------|-------|
| Number of classes | 58 |
| Min number of images per class | 10 |
| Max number of images per class | 119 |
| Mean number of images per class | 31 |
| Median number of images per class | 25 |
| Image file format | .jpg |
| Image dimension (width × height) | 800 px × variable |
| Number of annotated images | 1858 |
| *of which from Henrard* | 99 |
| *of which from Lapie* | 40 |
| *of which from Combier* | 7 |
| *of which from MRU* | 299 |
| *of which from Photothèque* | 711 |
| *of which from Internet* | 702 |
| Number of distractors | 11,316 |
| *of which from Henrard* | 935 |
| *of which from Lapie* | 4508 |
| *of which from Combier* | 1969 |
| *of which from MRU* | 2193 |
| *of which from Photothèque* | 1260 |
| *of which from Internet* | 451 |

Attributes take multiple values, two or three depending on the variation. We annotated the following variations with their respective possible values:

- Scale (what portion of the image does the object occupy?): Very close/Close/Midrange/Far
- Illumination: Under-illuminated/Well-illuminated/Over-illuminated
- Vertical orientation: Ground level (or street-view)/Oblique/Vertical
- Level of occlusion (is the object hidden behind other objects?): No occlusion/Partially hidden/Hidden
- Alterations (is the image degraded?): No alteration/Mildly degraded/Degraded
- Color: Color/Grayscale/Monochrome (e.g., sepia)

See Figure 2 for some examples of attribute annotations. In the full dataset, the images were picked to give an attribute distribution as uniform as possible, priorizing vertical orientation which we suspect to be the most influential; however, for some variations (illumination, occlusion, alterations), the distribution stays skewed towards a dominant value.

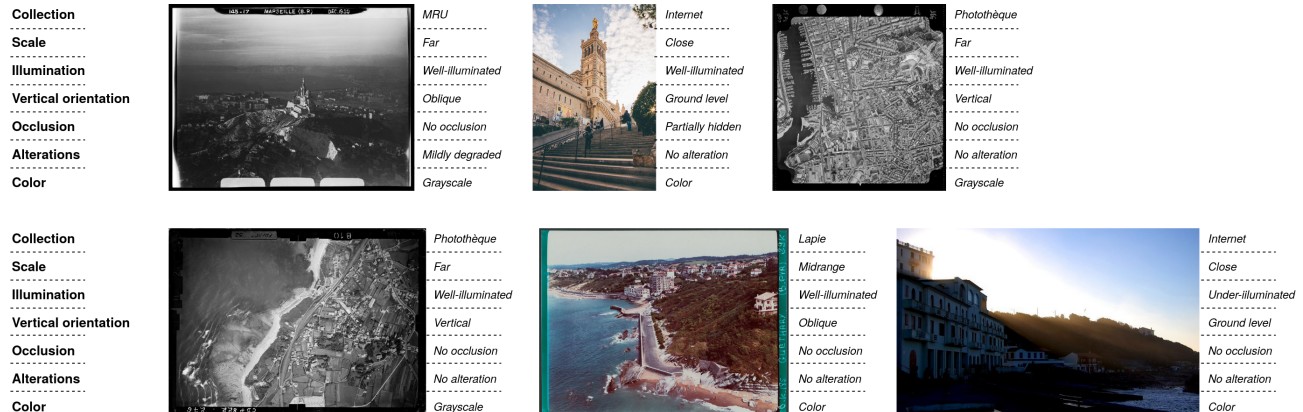

**Figure 2.** Examples from the ALEGORIA dataset, images in the same row belong to the same class (first row: "Notre-Dame de la Garde", second row: "Guethary"). Attribute values are indicated on the right side of images in *italics*, with attribute names in the legend at the beginning of each row.

On top of these attributes, we also include the source collection for each image, with the six possible collections: MRU, Lapie, Photothèque, Internet, Henrard, and Combier. Note that the last collection, Combier, does not contain enough images (7) to allow specific measures. Table 1 details the number of images for each collection.

Compared to the first version of the dataset presented in our previous work, this new version has more annotated images, more classes, and more precise annotations for the variations. Another key difference is the level of intraclass variation: the classes in our first version were mainly built with batches of images of the same location coming from the same source. In this version, we manually searched for all images of a given location in all of our sources, which drastically augment intraclass variation and thus the difficulty.

Figure 2 shows some examples from two different classes. Note the extreme scale, orientation, illumination, image ratio, and color variations. Even to the human eye, it is sometimes close to impossible to pair images from the same class without a certain knowledge of the context (here the hill and neighborhood of the cathedral), meticulous inspection of local patterns (in the fifth image, the cathedral is only visible in the distant background) and robustness to visual perturbations (in the third image, the cathedral is hidden behind film annotations). The dataset is voluntarily made challenging to exemplify real world situations of heterogeneous content-based image retrieval.

We use the term "heterogeneous" to emphasize the high variance in object representations, but we note that the literature generally formulates this type of problem with the "domain view". Domains can be arbitrarily defined and could be used here to refer to different cases along a given variation (vertical orientation), with each attribute value defining a domain. To be coherent with the literature, we will consider that a given collection is specific enough to define a domain, and we will therefore use the terms **domain** and **collection** interchangeably in the following. However, we invite the reader to keep in mind that this is an arbitrary semantic definition, and that it does not accurately represent all underlying visual characteristics.

### 2.1.2. Cultural Heritage Datasets

A recent review [9] gives an overview of datasets for cultural heritage, most of them being datasets of artworks (paintings, drawings). While our sources are the same (GLAMs), our task differs by its semantics and applications. A close task to ALEGORIA is the Arran benchmark [10], where LiDAR data are used to detect objects in archeological sites.

If we broaden the notion of cultural heritage beyond artworks, more datasets can be found, such as the HistAerial dataset [11], which proposes 4.9M patches extracted from vertical aerial imagery acquired between 1970 and 1990, with land-use annotations; or the Large Time Lags Locations (LTLL) [12] dataset, with 500 images of landmarks in a cross-domain setup of old photography versus modern pictures, all taken from the

ground. Similarly, the commonly used Oxford5k [13] and Paris6k [14] benchmarks for image retrieval (on which unprecedented performance has been obtained thanks to the large GoogleLandmarks [15] training dataset, later presented in Section 2.3.1) focus on landmarks and are thus valuable sources of data representing our past, but only include street-level photography.

With the ALEGORIA benchmark, we propose to generalize further: 1. by mixing old and new photography (as in LTLL), 2. by mixing aerial, oblique and ground photography (as in University1652, later presented in Section 2.3.1), 3. without restraining classes to easily identifiable objects (as in Arran or HistAerial), 4. in a challenging setup suited for evaluating modern deep learning-based retrieval methods, with dedicated evaluation protocols and measures (as in Oxford5k and Paris6k).

We note that, in a similar trend, there has been rising interest for matching images in long-term or cross-season setups, with applications notably in autonomous driving, but we will not mention these works here as they treat a different problem (retrieving only one positive image is sufficient) with different data (ground-level road sceneries from embedded systems).

### 2.1.3. Evaluation Protocol

Retrieval performance on the ALEGORIA benchmark is evaluated along three setups:

- Absolute retrieval performance: the average quality of results lists obtained when using all annotated images as queries, regardless of domain considerations.
- Intra-domain or attribute-specific performance: the retrieval performance obtained when using a subset of annotated images from a specific collection or with a specific attribute value. This allows a finer comparison along different representation domains and characteristics.
- Inter-domain performance: the ability to retrieve images outside of the query domain.

**Absolute retrieval performance** is measured with the mean Average Precision (mAP). Following the notation proposed by [16], the Average Precision for query $q$ is defined as:

$$AP_q(P, \Omega) = \frac{1}{|P|} \sum_{i \in P} \frac{R(i, P)}{R(i, \Omega)} \tag{1}$$

where $P$ is the set of positive images for query $q$, $\Omega = P \cup N$ is the set of all images (positives $P$ and negatives $N$), $R(i, P)$ is the ranking of image $i$ in $P$, and $R(i, \Omega)$ is the ranking of image $i$ in $\Omega$. Rankings are obtained by sorting pairwise image similarity scores (which depend on the descriptor used) in decreasing order. The mAP is computed by averaging APs over the set $Q$ of 1858 queries. Sets $P$ and $N$ are usually the same for all images belonging to the same class. However, in the ALEGORIA dataset, some images show objects from multiple classes. In these cases, $P$ is specific to the query and includes all images containing one of the objects of interest.

**Intra-domain performance** is measured with mAP scores using the provided collection and representation attributes, on a subset of the queries. The collection- or attribute-specific mAP is defined as the mAP obtained when filtering $Q$ with collection or attribute values, respectively. For example, the intra-domain performance for the collection "Lapie" is measured as the mAP computed on the subset of 262 Lapie queries. Similarly, the intra-domain performance for the attribute "Scale", value "Very close" is measured as the mAP computed on the subset of 242 queries with this value. Note that these various mAP scores are all computed on different sets of queries and therefore should not be compared to each other; they only allow comparison of different methods on the same setup.

**Inter-domain performance** is measured with a set of statistics based on the position of positive images from different collections. We propose two indicators based on P1, the position of the first positive image from a different collection (see Figure 3 for a visual representation of P1): median P1 (mP1) and the first quartile of P1 (qP1). These two measures give a rough idea of how positive images from different collections are

distributed in the list of results. We also compute the mean Average Position Deviation (mAPD) as the difference between the average position of positive images from different collections and the average position of all positive images (we compute this deviation for all queries and get the mean value, similarly to the mAP). This measure has the advantage of not depending on the number of positive images and the absolute performance of the method, and is easily interpretable: the ideal value is zero (images are retrieved regardless of their collection, i.e., their average position stays the same), and higher values indicate that positive images from different collections are further in the list of results.

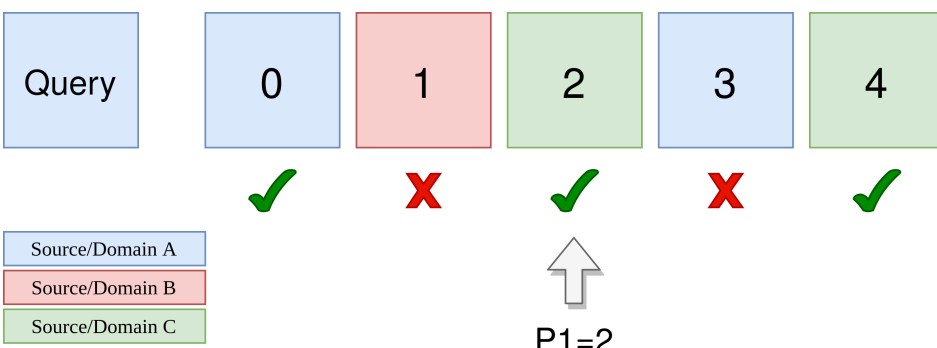

**Figure 3.** Calculation of P1, the position of the first positive image from a different domain (or collection), for a single query. Different colors indicate different domains, True/False symbols indicate if the corresponding image is positive or not for this query.

### 2.2. Retrieval Framework

Figure 4 shows the usual framework for content-based image retrieval. The separation between offline and online processes makes it particularly suited for handling cultural heritage: after the first index build, images can be appended to it whenever necessary (when a new digitized batch is ready for example), while allowing image search at any moment. It is also compatible with multimodal retrieval where indexes built with text metadata or other modalities can be combined.

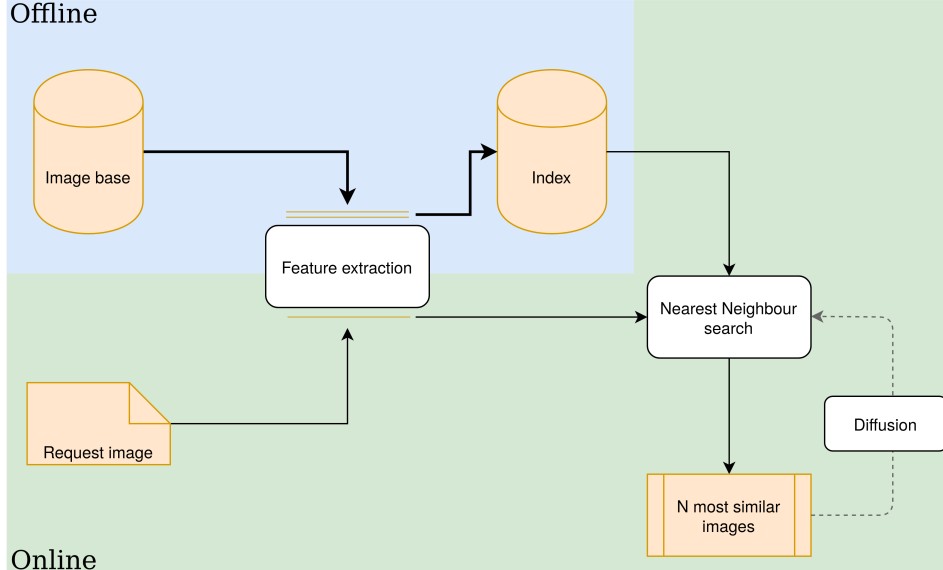

**Figure 4.** Schematic Content-Based Image Retrieval (CBIR) framework. The offline part computes descriptors for the whole image base and stores them in the index. The online part, using the same feature extractor, computes descriptor(s) for the query and retrieves the N most similar images using a similarity measure to get the N nearest neighbors in the index. An additional diffusion step may be used to refine the initial results.

In this paper, we focus on two computational steps that we consider most important for cross-domain retrieval: Feature extraction and Diffusion. We believe that these two axes are the tools to balance absolute, intra-domain, and inter-domain retrieval performance.

**Feature extraction** refers to the process of computing one or multiple descriptors per image. Deep convolutional neural networks (CNNs) are now commonly used as feature extractors, yielding results superior to hand-crafted features [1,17] due to their ability to adapt to a given task and their high expressivity. Using a backbone CNN, feature extraction consists of passing an image through a series of layers to get a tensor of activations (a 3D volume of high-level information) which is processed to extract local or global descriptors (or both [18]). See Figure 5 for the principle of deep descriptors. Global descriptors vary depending on how they process the tensor of activations: simple pooling operations such as sum [19], max [20], or generalized mean (GeM) [21]; or more complex operations such as cross-dimensional weighting [22] or second-order attention maps [23]. Local descriptors rely on a selection operation: the tensor of activations is considered as a set of local descriptors, which must be filtered with attention mechanisms to only keep the most discriminative ones. Noh et al. [15] developed the first large-scale retrieval system with local descriptors, later generalized in [24]. Apart from architecture choices, the training process also plays a major role, and in particular sampling (how training images are picked) and the loss function used to optimize the network. These complementary choices will be discussed in Section 2.3.1.

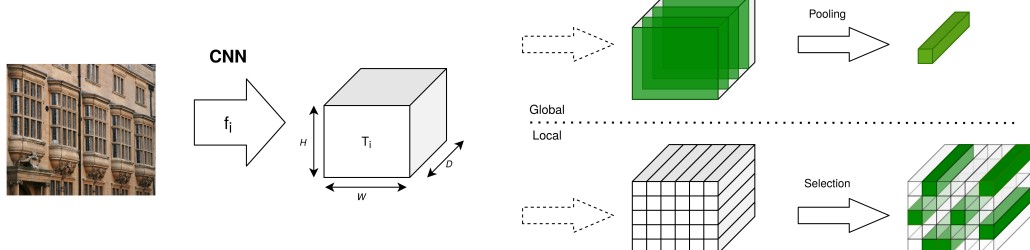

**Figure 5.** Deep descriptor extraction principle. The tensor *Ti* of activations can be seen as a C-dimensional stack of activation maps that can be reduced to a single C-dimensional global vector, or as an array of C-dimensional local vectors that must be filtered to keep the N most discriminative local descriptors.

**Diffusion** is an optional post-processing step consisting of exploiting information from the database to enhance the list of results. Concretely, this can done by using the first *N* results of the initial search to re-compute descriptors (a process also referred to as Query Expansion-QE). Early methods simply combined the query descriptor with the descriptors of the first results by averaging them [19,20,22,25], and have been improved later with $\alpha$-weighted query expansion where similarities are used as weights when averaging [21]. These approaches have the benefit of staying relatively economic in computational overhead, while bringing significative improvements in many setups. More advanced methods go further by exploiting the affinity matrix, defined as the set of pairwise similarities between a given image and all other images. For a database $\Omega$, a similarity matrix *S* is calculated with pairwise similarities:

$$s_{i,j} = sim(d(x_i), d(x_j)), \forall (x_i, x_j) \in \Omega \tag{2}$$

*sim* can be any measure of similarity, depending on the considered descriptor *d* (cosine similarity, Euclidean distance on global descriptors, matching kernels [26] on local descriptors...). This matrix can be interpreted as a graph [27], where each image is a node, and edges linking nodes to each other are weighted using the corresponding similarity $S_{ij}$. In its simplest version, the adjacency matrix is all ones since each image is connected to each other image:

$$a_{i,j} = 1, \forall (i,j) \in [1..|\Omega|]^2 \tag{3}$$

Diffusion is conducted by updating node features or directly $S$ using an update rule. Donoser and Bischof [28] presents a set of strategies regarding this update rule. This modular setup allows offline computation [29] and region-based variations [30]. Note, however, that using all neighbors ($A$ from Equation (3)) is computationally expensive considering the $|\Omega|$ node updates depending on the $|\Omega|$ adjacent nodes. A simple but effective way to avoid this systematic computation is to filter edges to only keep the $k$ nearest neighbors:

$$a_{i,j} = \begin{cases} 1, & \text{if } x_j \in NN_k(x_i), \\ 0, & \text{otherwise} \end{cases} \tag{4}$$

with $NN_k(x_i)$ the $k$ nearest neighbors of $x_i$ according to $S$. Zhong et al. [31] and Zhang et al. [27] show good results with a similar setup going further using reciprocal nearest neighbors.

### 2.3. The Low-Data Problem

In this section, we present some methods to handle the first problematic domain gap on the ALEGORIA problem: the gap between currently available training data and ALEGORIA images. Indeed, the low volume of images in the benchmark does not make it suited for training CNNs.

The goal of uncoupling test performance from training data has given rise to a variety of research topics that can be sorted as shown in Figure 6 along the supervision axis.

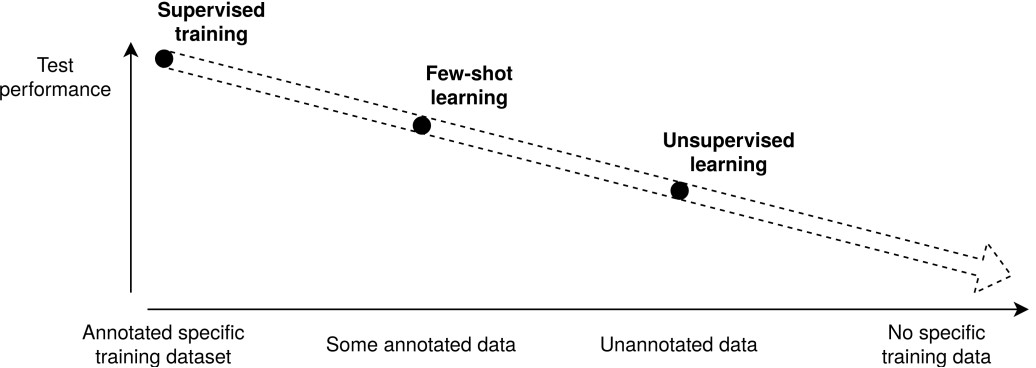

**Figure 6.** Overview of research topics along availability of training data. Typical learning setups differ according to their data-dependency (*x*-axis), with expected test performance (*y*-axis) decreasing with less training data.

### 2.3.1. Supervised Learning or Transfer Learning: Fine-Tuning on Related Datasets

Supervised learning corresponds to optimizing the backbone on a setup (characterized with a task and an associated dataset ) as close as possible to the target setup, using large volumes of annotated data. Annotations typically take the form of a single class label associated with an image as in the ImageNet dataset [32] but can also be more detailed as the bounding boxes in [33] or derived from 3D models [21]. Two aspects of supervised learning that are essential in our problem are the choice of the training dataset and the loss function:

**The training dataset** will largely affect the behavior of the descriptors. Fine-tuning a feature extractor on a training dataset related to the target dataset has been proved to be crucial [21,34,35]. However, how can we pick such a dataset? Table 2 compares some datasets that might be considered regarding ALEGORIA.

**Table 2.** Overview of available training datasets.

| Training Dataset | Semantics | Number of Classes | Size |
| --- | --- | --- | --- |
| Imagenet [32] | Generalistic | 1000 | 1.3M |
| GoogleLandmarks [15] | Landmarks | 81k | 1.4M |
| SF300 [36] | Remote Sensing | 27k | 308k |
| University1652 [37] | Cross-view buildings | 701 | 50k |
| SfM-120k [21] | Landmarks | 551 | 120k |
| Landmarks [38] | Landmarks | 672 | 214k |

**The loss function** used to optimize the backbone is also a matter of debate. Image retrieval is a task close to classification, in the sense that it implies learning discriminative features to separate classes. The main difference lies in the available information regarding classes: classification assumes a fixed number of known classes, while image retrieval is applied to an unknown number of undefined classes. Nonetheless, works using the standard cross-entropy loss [15] or one of its enhanced version, the ArcFace loss [18], both designed for classification, have shown satisfying results in retrieval. Another approach consists of sampling a positive and negative pair of images, and optimizing to pull the former closer and the latter further apart, with a triplet [34] or a contrastive loss [21]. We also note a recent trend in directly optimizing with a differentiable approximation of the evaluation metric (the mean average precision) [16,39]. This is arguably more consistent with the testing setup of image retrieval, but, in reality, results do not show a clear difference with classification or tuplet losses [33].

Setting technical questions apart, we note that the supervised learning setup is very limited in its possibilities: the assumption that a suitable training dataset will be available for every application is difficult to meet. However, and considering the state of the art in "less"-supervised setups, it is still the most straightforward way to approach a new application.

### 2.3.2. Few-Shot Learning, Meta-Learning: A Promising Path?

Starting from the conclusions of Section 2.3.1, a recent branch of research is investigating how to learn using a limited number of annotated data, or more broadly how to learn more efficiently (learn to learn). Wang et al. [40] presents a taxonomy of few-shot learning methods separating approaches exploiting prior knowledge in the data, in the model or in the algorithm. Few-shot learning is usually formulated with episodic training. An episode is formed with a query set, on which we want to conduct a task (e.g., classifying cats and dogs), and a small support set that contains valuable information regarding the query set (e.g., examples of cats and dogs). The model is optimized through episodes, with the goal of making it learn how to rapidly gain accuracy on the query set using a limited support set.

We selected one promising approach: Conditional Neural Adaptive ProcesseS (CNAPS) [41,42] inserts Feature-wise Linear Modulation (FiLM) [43] layers in a backbone feature extractor for fast adaptation. Using episodic training and a deterministic Mahalanobis distance, the improved version SimpleCNAPS [42] trains an adaptation network (without updating the backbone) to produce adapted FiLM parameters and obtains good performance on the MetaDataset [44] benchmark. See Figure 7 for the principle of the FiLM layer in few-shot learning.

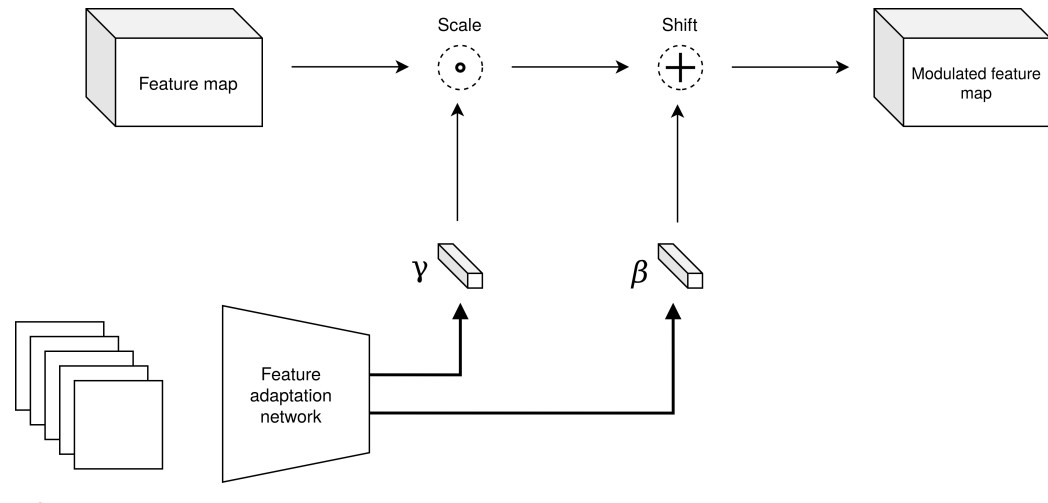

**Figure 7.** Principle of a FiLM layer applied to few-shot learning: the support set passes through a feature adaptation network, producing $\gamma$ and $\beta$ parameters that will respectively scale (the dot represents the Hadamard product, or channel-wise multiplication) and shift (the cross represents channel-wise addition) feature maps at different levels of a convolutional neural network.

Note that, while many interesting contributions have been made for few-shot learning, there are still significant obstacles to real world applications, including image retrieval:

- To our knowledge, there has been no work applying the principles of few shot learning to image retrieval (rather, few shot learning has borrowed ideas from image retrieval [45]). It is indeed a tricky situation in the sense that image retrieval does not use class information (except when evaluating), and thus defining a support set is not trivial.
- Most of the existing works have been tested on simplistic datasets with low resolution, few problematic variations, and semantically easy classes, e.g., the standard MetaDataset [44] grouping ImageNet, Omniglot (character recognition) [46], VGG Flowers (flower recognition) [47], Traffic Signs (traffic signs recognition) [48], and other relatively simple datasets.
- Due to the computational overhead induced with meta-learning architectures, small feature extractors such as ResNet-18 or its even smaller version ResNet-12 are used [49] to avoid memory limitations. This naturally also limits final performance.
- Simple baselines get better results than complex meta-algorithms [49] on some setups, which indicates that there is still room for improvement.

### 2.3.3. Unsupervised and Self-Supervised Learning

Concurrently to approaches working with limited annotated data, other works have developed strategies to learn without any annotation. Unsupervised and self-supervised learning (one might argue that self-supervision is a form of unsupervised learning where intermediate generated labels are used, but we will use both terms interchangeably here) can only rely on automated discovery of patterns in the data to conduct the given task. Some ideas with encouraging results include:

- Teaching a model how to solve Jigsaw puzzles [50] generated by selecting patches of an image, to automatically learn the important parts of an object. This is an example of a pretext task (solving puzzles) making the model learn semantically important features (shapes, relative spatial positions) that can be reused for a downstream task (in our case retrieval).

- Learning image generation models with the Generative Adversarial Networks architecture [51]. In this setup, a generative model competes with a discriminative model. The generative model tries to fool the discriminator by producing realistic fake images, while the discriminator tries to distinguish fake images from real images. Here, the discriminator provides a form of automated supervision to the generator, using only pixel data from a database of images. By reconstructing realistic images, the generative model is forced to get a visual understanding of the object. Applications to discriminative tasks have shown that the learned features do contain discriminative information [52], but for now only on the very basic MNIST [53] dataset.
- Leveraging data augmentation techniques to learn visual patterns without labels. Recall the tuplet losses presented in Section 2.3.1: the positive pairs that we pull together (while we push negative pairs apart) can be automatically generated with a single image on which we apply simple transformations such as cropping, distortion, blur... Chen et al. [54] achieved results similar to early supervised models on ImageNet classification with this framework.

Lastly, and considering our task of image retrieval, note that the handcrafted local descriptors such as SIFT [55] or ORB [56] that were used before deep learning methods do not require any form of supervision (nor any learning). Combined with advanced detectors and indexing methods, they can provide competitive results [17], but we will not detail them further here following early evaluations showing a clear advantage of deep local and global descriptors [8], and because they have less room for improvement.

### 2.4. The Heterogeneity Problem

The second problematic domain gap on the ALEGORIA problem comes from the high variance in representation characteristics. The keywords we used to browse the literature tackling this issue were "domain generalization", "cross-domain matching/retrieval", "invariance"... We note that most of our readings either approach the problem as a whole, proposing solutions that are application-agnostic but with low performance, or only consider a specific problem such as cross-view matching, with satisfying performance but untransferable methods. Moreover, few of these techniques were put into the context of image retrieval, where compact descriptors are expected to allow scalability. The methods we will present in this section are thus strictly in the context of robust feature learning, to avoid presenting techniques that could not be applied to image retrieval.

#### 2.4.1. Reducing Variance

A first and easy path to consider is to "manually" reduce heterogeneity. Notably on ALEGORIA, three sources of variation, observable in Figure 2, can be handled with preprocessing transformations: color, visual noise (film borders, vignetting) and varying scales. We include some experiments in Section 3 to study if such transformations can help in matching images.

Manual transformations can also be applied during training, to emulate variance in training data. This virtually augments the volume of training data, thus the name data augmentation. Simple transformations such as cropping, blurring, and jittering have been proven to be beneficial to learn more robust models [57], and can even be sufficient to learn representations, as shown by SimCLR [54], which only relies on basic image transformations. It is now a common practice to use a relevant panel of random image transformations when sampling training images. State-of-the-art descriptors presented in Section 2.5 as well as our own implementations all used a form of data augmentation during training. We will, however, not further elaborate on these transformations since the ALEGORIA benchmark is characterized by complex transformations (such as vertical orientation changes) that can not be manually emulated.

Following the recent success of Generative Adversarial Networks (GANs) [51] for realistic image generation and modification, some works have proposed to further elaborate on data augmentation by generating more complex image transformations using deep

networks, with satisfying results in specific contexts [58,59]. However, some important caveats remain before applying such techniques for training generalistic robust descriptors: GANs are hard to train [60] and need to be constrained to generate relevant samples while avoiding altering discriminative information [61]. It is still unclear if building and training a GAN to augment a dataset with generated samples is better than directly training a discriminative model on all the available training data.

### 2.4.2. Robust Feature Learning

In the context of image retrieval, robustness has always been a desirable quality of descriptors. Note, for example, how traditionnal hand-crafted descriptors such as SIFT [55] or SURF [62] directly integrated robustness in their formula through operations mathematically invariant to local changes. In the deep learning era, such properties can be integrated either through the architecture (i.e., the operations done on the input image) or through optimization (i.e., by constraining parameter space with a loss function). Detector TILDE [63] and descriptor LIFT [64], for example, enforced invariance properties on local descriptors through careful architectural and loss design. However, such methods lack semantic interpretation and thus have limited accuracy, descriptors trained only on discriminative objectives getting better results [15] without using any tailored method for robustness.

Some propositions have explored how to learn robust features through two distinct domains, e.g., cross-view matching [65], night-to-day matching [66], or any situation where training pairs are available [67]. Zheng et al. [37] in particular proposes a baseline for learning descriptors robust to vertical orientation changes.

### 2.5. Contributions and Experiments

Using our benchmarking setup evaluating 1. absolute performance, 2. intra-domain performance, and 3. inter-domain performance, and considering the panel of ideas presented in Sections 2.3 and 2.4, we conducted the following experiments using available methods:

Unsupervised learning (**SimCLR**): using a reimplementation of the original method (see Section 2.3.3 with ResNet50 as the feature extractor (we modified the last layer to produce 512-dimensional descriptors), we train the model on the ALEGORIA distractor set (11k images) until convergence, and test the produced descriptors on ALEGORIA.

Pretrained (**GeM**): using ResNet50 trained on ImageNet as a backbone feature extractor (weights provided by common deep learning libraries) and GeM pooling, we produce 2048-dimensional global descriptors.

Fine-tuning (**GeM-ArcFace**): using ResNet50 as a backbone feature extractor (with a linear dimension reduction layer at the end bringing final descriptor dimension to 512), GeM pooling, two training datasets presented in Table 2, and the ArcFace loss, we fine-tuned the model weights, stopping when validation accuracy reached a maximum.

State-of-the-art methods (**\*[Desc]-[Loss]**: using the weights provided by authors, we test four state-of-the-art methods: GeM trained with the Triplet Loss on the multi-view University-1652 dataset [37], GeM trained with the Contrastive Loss on SfM120k [21], RMAC trained with the AP ranking loss on Landmarks [39], and the local descriptor HOW trained with the Contrastive Loss on GoogleLandmarks [24]. Global descriptors are compared with the cosine similarity, while HOW relies on the ASMK matching kernel [26] to compute pairwise similarities.

Alpha Query Expansion (**$\alpha$QE**): from a kNN adjacency matrix (see Equation (4)), we update node features using an exponentially weighted update rule (single update):

$$d_{x_i} = \sum_{x_j \in NN_k(x_i)} (s_{i,j})^\alpha * d_{x_j} \qquad (5)$$

### 2.5.1. Fast Adaptation

We propose to investigate how a few-shot learning method, CNAPS (see Section 2.3.2), can be adapted for low-data image retrieval scenarios. Using the same networks as [41] (ResNet18 with FiLM layers inserted, set encoder, adaptation network), we make the following modifications:

We adopt the Smooth-AP loss [16], which allows for directly optimizing a retrieval objective instead of a classification proxy objective. During training, we randomly select $N$ classes, sample 5 images per class, and concatenate to build mini-batches of $5N$ images.

We reformulate episodic training: in traditional few-shot learning, other images from the same class are used. After experimenting with different setups, we got the best results using the self-supervised setup: during training, the support set is the same as the query set.

We systematically evaluated training setups with a panel of test setups:

A　　Use one random image from the ALEGORIA benchmark as support to compute adapted parameters, then extract all descriptors with the same adapted parameters.

B　　Same as A but with five random images.

C　　For each image, we first compute adapted parameters using the same image, and then extract its descriptor. Descriptors of different images are thus extracted with different parameters.

D　　We first issue an unadapted image search with the backbone extractor. Using the first retrieved image as the support set, we compute adapted parameters (different for each image), extract descriptors, and conduct another search.

E　　For comparison, we also include an experiment with noise as support: we compute adapted parameters with random values sampled from the normal distribution as input, and extract descriptors with these parameters (same for all images).

We will refer to this method as **rCNAPS** for retrieval-adapted CNAPS.

### 2.5.2. Multi-Descriptor Diffusion: Balancing Intra- and Inter-Domain Performance

Following Equation (2) and the graph diffusion framework, we propose to explore how to generalize to multiple descriptors, i.e., how to build and propagate on a graph from multiple similarity matrices $S^1$, $S^2$, $S^3$...; with the motivation of exploiting the different properties of different descriptors.

If $S^v$ is the similarity matrix issued from the $v$-th descriptor, and we have Y descriptors, we first build Y kNN graphs:

$$a_{i,j}^v = \begin{cases} 1 & \text{if } x_j \in NN_{k1}^v(x_i), \\ 0 & \text{otherwise} \end{cases} \tag{6}$$

with $NN_{k1}^v(x_i)$ the $k1$ nearest neighbors of $x_i$ according to $S^v$. Note that $k1$ stays the same across graphs, to avoid multiplying parameters and because it ideally corresponds to the average number of positive images for a query, a value independent from the description method. Note that kNN graph building is independent from the descriptor used: the only input is a similarity matrix that can be obtained from a cosine similarity, a Euclidean distance, a matching kernel between local descriptors, or any method outputting pairwise similarities.

Following [27], we make the graph reciprocal (i.e., $A$ symmetrical), which can be done in a simple operation:

$$A^* = \frac{A + A^T}{2} \tag{7}$$

This way, we get:

$$a_{i,j}^v = \begin{cases} 1 & \text{if } x_j \in NN_{k1}^v(x_i) \wedge x_i \in NN_{k1}^v(x_j), \\ 0.5 & \text{if } x_j \in NN_{k1}^v(x_i) \vee x_i \in NN_{k1}^v(x_j), \\ 0 & \text{otherwise} \end{cases} \tag{8}$$

We conduct a first diffusion step to refine descriptor-specific similarities. As in [27], we propagate only using the top $k2$ neighbors:

$$s_i^v \leftarrow \sum_{j \in NN_{k2}^v(x_i)} a_{i,j}^v * (s_{i,j}^v)^\alpha * s_j \tag{9}$$

This is an exponentially weighted update rule similar to $\alpha$QE (Equation (5)): the vector $s_i^v$ of similarities between image $i$ and other images is updated by a weighted sum of neighboring similarity vectors. Note that here we allow matrix $A$ to take non-integer values, which differs from the standard graph notation where $A$ is only zeros and ones, but this stays compatible with the graph view if we consider that edges contain both similarities and an additional weight in $\{0.5, 1\}$. To summarize, for each image, we use the top $k1$ neighbors to define a feature vector based on similarities, and we refine this feature vector with an update based on the top $k2$ neighbors.

Matrices $S^v$ are L2-normalized, and merged:

$$S = \frac{1}{Y} \sum_{v \in [1..Y]} S^v \tag{10}$$

From this step, we follow the same logic again with merged similarities. We recompute $A$ using similarities instead of descriptors as features (see Equation (4), we keep the same $k1$), and we update similarities with the top $k2$ neighboring nodes:

$$s_i \leftarrow \sum_{j \in NN_{k2}(x_i)} a_{i,j} * (s_{i,j})^\alpha * s_j \tag{11}$$

Using the final similarity matrix $S$, we evaluate on ALEGORIA. We coin this method Multidescriptor Diffusion (**MD**).

We also propose an alternative method introducing annotation information in the diffusion process: in $A$, we can force certain values based on external criteria. Specifically, using the domain annotations provided in the ALEGORIA benchmark, we can define an inter-domain matrix $T$:

$$t_{i,j} = \begin{cases} 1 & \text{if } domain(x_i) \mathrel{!=} domain(x_j), \\ 0 & \text{if } domain(x_i) = domain(x_j) \end{cases} \tag{12}$$

This allows us to force connections between images from different collections (positive or negative, we obviously do not use class annotation). There is intuitively a compromise to solve here between descriptor-specific performance and out-of-domain retrieval, and we thus introduce a $\lambda$ parameter to allow tuning. After Equation (8), we merge $T$ into $A$ with:

$$A \leftarrow A + \lambda * T \tag{13}$$

We will refer to this method as constrained Multidescriptor Diffusion (**cMD**).

## 3. Evaluations and Results

This section presents and discusses several experiments conducted to evaluate the performance of the main state-of-the-art approaches on ALEGORIA along different test setups. Table 3 summarizes the results of the considered methods.

**Table 3.** Performance comparison of various descriptors on ALEGORIA, with absolute, intra-domain, and inter-domain measures. Descriptors with a star (*) are extracted using authors' provided model weights; other descriptors are from our own reimplementation and training. Best performance for each measure (column) is in **bold**. Absolute and intra-domain performance is measured with the mAP (the higher, the better). Inter-domain performance is measured with specific indicators detailed in Section 2.1.3, mP1 and qP1 do not have optimal values, while optimal mAPD is zero (the lower, the better).

| Method | Training Dataset | Reranking | Absolute Perf. (mAP) | | Intra-Domain Performance (mAP) | | | | | Inter-Domain Performance | | |
|---|---|---|---|---|---|---|---|---|---|---|---|---|
| | | | ALEGORIA | MRU | Lapie | Photothèque | Internet | Henrard | mP1 | qP1 | mAPD |
| Unsupervised | | | | | | | | | | | |
| **SimCLR** | ALEGORIA distractors | | 5.77 | 10.22 | 7.89 | 5.01 | 4.29 | 7.66 | 134 | 40 | **176.8** |
| Semi-supervised | | | | | | | | | | | |
| **rCNAPS** | GoogleLandmarks | | 8.47 | 11.02 | 2.91 | 8.2 | 7.82 | 9.88 | 101 | 24 | 237.1 |
| Supervised | | | | | | | | | | | |
| ***GeM** | ImageNet | | 14.25 | 23.63 | 19.92 | 11.28 | 12.98 | 13.93 | 152 | 23 | 290.9 |
| **GeM-ArcFace** | GoogleLandmarks | | **24.30** | 27.16 | **29.43** | 13.45 | **34.10** | **20.33** | 67 | 10 | 251.0 |
| **GeM-ArcFace** | SF300 | | 14.41 | 26.27 | 17.76 | 14.44 | 9.33 | 13.32 | 99 | 20 | 256.5 |
| ***GeM-Triplet** | Univ1652 | | 10.02 | 20.33 | 15.37 | 8.98 | 6.45 | 10.17 | 100 | 25 | 239.1 |
| ***GeM-Contrastive** | SfM120k | | 19.02 | 26.39 | 19.70 | **14.59** | 20.57 | 16.83 | 107 | 18 | 281.1 |
| ***RMAC-APLoss** | Landmarks | | 19.97 | 24.82 | 20.25 | 13.96 | 24.38 | 15.49 | 79 | 13 | 261.5 |
| ***HOW-Contrastive** | GoogleLandmarks | | 19.16 | **28.11** | 19.53 | 10.20 | 24.45 | 17.55 | 105 | 21 | 274.0 |
| Diffusion | | | | | | | | | | | |
| **GeM-ArcFace** | GoogleLandmarks | + αQE | 25.02 | 27.95 | 30.41 | 13.19 | 35.81 | 20.57 | 77 | 11 | 250.6 |
| **GeM-ArcFace** | GoogleLandmarks | +GQE | 27.41 | 30.60 | 45.08 | 15.01 | 37.82 | 23.94 | 78 | 11 | 284.8 |
| **GeM-ArcFace** **GeM-Contrastive** **HOW-Contrastive** | GoogleLandmarks SfM120k GoogleLandmarks | +MD | 29.17 | 35.28 | 39.70 | 17.92 | 37.53 | 26.00 | 97 | 13 | 327.5 |
| **GeM-ArcFace** ***GeM-Contrastive** ***HOW-Contrastive** | GoogleLandmarks SfM120k GoogleLandmarks | +cMD | 29.10 | 35.31 | 41.40 | 17.87 | 37.35 | 25.60 | 81 | 12 | 279.5 |

### 3.1. rCNAPS Experiments

We conducted some experiments stated in Table 4 to validate our adapted implementation of CNAPS.

**Table 4.** Validation experiments for rCNAPS on ALEGORIA.

| Evaluation Setup | Absolute Perf. (mAP) |
|---|---|
| Unadapted | 5.81 |
| A (few-shot, k = 1) | 8.34 |
| B (few-shot, k = 5) | **8.47** |
| C (self-supervised) | 7.91 |
| D (query expansion, k = 1) | 5.72 |
| E (noise) | 8.00 |

Note that using noise already gives a significant performance boost compared to an undapated feature extractor. This is due to the additional expressivity brought by the set encoder and adaptation networks, optimized through training with a retrieval objective.

Setups C and D do not surpass noise. We hypothesize that this is due to the different adapted parameters used to extract descriptors, and to the high variance of the ALEGORIA benchmark: the additional information brought by support images do not compensate the noise generated by the varying feature extractor parameters.

Surprisingly, we obtain the best results with setup B, which does not confirm the usual guideline that the training setup must match the testing setup in meta-learning. We hypothesize that setup B is the more robust to variations, thanks to the set encoder that filters out uninformative support images by pooling through the five available images. We include results obtained with setup B in Table 3 to put them in context with methods

using less or more annotations. It should be noted that these experiments are done as an exploratory proposition and can certainly be enhanced, but this is not our objective here.

### 3.2. Preprocessing

As presented in Section 2.4.1, simple transformations applied to test images can already help us gain some performance. Table 5 shows the effect of inserting two simple preprocessing steps with virtually no computational overhead: switching all images to grayscale, and cropping images with a 0.9 ratio. Motivated by the small gain in accuracy, we apply this preprocessing for all experiments in Table 3.

**Table 5.** Effect of preprocessing test images on absolute performance, with GeM trained on Google-Landmarks.

| Grayscale | Crop | Absolute Perf. (mAP) |
|:---:|:---:|:---:|
| | | 23.68 |
| ✓ | | 23.78 |
| | ✓ | 24.17 |
| ✓ | ✓ | 24.30 |

## 4. Discussion

### 4.1. Supervision Axis

Experiments with SimCLR, rCNAPS and the panel of fully-supervised methods confirm our expected drop in performance from Figure 6. Even if the training datasets from supervised methods do not exactly match ALEGORIA statistics, their volume brings enough variety to compute descriptors with better accuracy and generalization ability. The global mAP with GeM trained on ImageNet, a generalistic dataset semantically very far from ALEGORIA, is ∼6 points higher than the mAP with rCNAPS, highlighting the importance of the training dataset.

### 4.2. Intra-Domain Performance Disparities

Separating absolute performance from intra-domain performance allows us to better understand underlying behaviors: some descriptors comparatively perform better on a domain, even if their absolute performance is lower. For example, descriptor HOW gives a better mAP on collection MRU than GeM-ArcFace trained on GoogleLandmarks, even if its absolute performance is ∼4 points lower.

We note that the notion of domain, as presented in the literature, stays relatively vague in its definition: it can be understood as any criterion that allows clustering of the data into separate groups. In our case, we use the term domain to refer to a collection because we are interested in matching images through different collections, but studying domain-specific performance does not inform us about the fundamental image variations that impact how descriptors behave. To better understand and visualize this, we use the available variation annotations to compute attribute-specific mAP depending on the training dataset, with GeM global descriptor and HOW local descriptor for comparison. Results are shown in Figure 8.

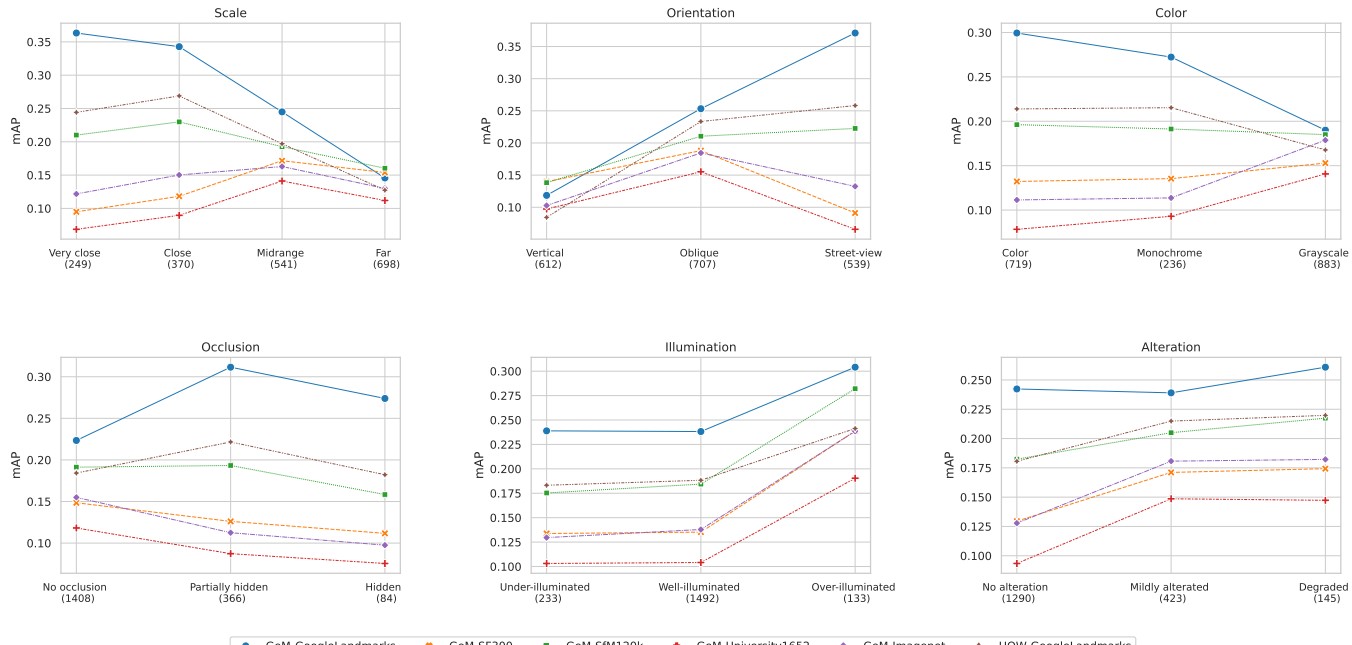

**Figure 8.** Attribute-specific performance evaluation. The number of corresponding query images is noted in parentheses. Performance should be compared between descriptors on a single value because the varying query set can make absolute performance change independently from attribute variations.

We notice that GeM-GoogleLandmarks are better than other descriptors in most cases, except a particular type of images: vertical images, with very small objects. This can be explained with the main semantic of GoogleLandmarks being street-level tourist photography.

### 4.3. Diffusion

The two state-of-the-art diffusion methods we tested, $\alpha$QE and GQE, improved performance by respectively 0.72 and 3.11 points of mAP. Our proposed multi-descriptor method using the three best performing descriptors in intra-domain performance brings an improvement of 4.87 points. The constrained version does not further improve absolute performance but changes inter-domain performance as we will detail later.

Optimal parameters in terms of absolute performance found for $\alpha$QE are $(\alpha, n) = (1, 3)$, for GQE $(k1, k2, \alpha) = (38, 5, 1.0)$; for MD $(k1, k2, \alpha) = (15, 4, 7)$ and for cMD $(k1, k2, \alpha, \lambda) = (17, 4, 9, 0.1)$.

Our proposed MD and cMD methods use respectively three and four hyper-parameters. To study how these parameters influence the performance, we first put aside cross-domain performance (with the $\lambda$ parameter of cMD) and evaluate the evolution of the absolute mAP when varying $k1$, $k2$, and $\alpha$. Figure 9 shows a heatmap of the absolute mAP against $k1$ and $k2$ ($k2 < k1$), with $\alpha$ fixed to 7. Apart from the obviously suboptimal region of $(k1, k2) < (3, 3)$, which does not exploit enough neighboring information, and a decreasing performance when reaching high values, there is a near-optimal zone of $(10, 3) < (k1, k2) < (22, 20)$ where absolute performance is stable regardless of varying $k1$ and $k2$. This indicates that tuning these parameters should not be a problem on cases similar to ALEGORIA.

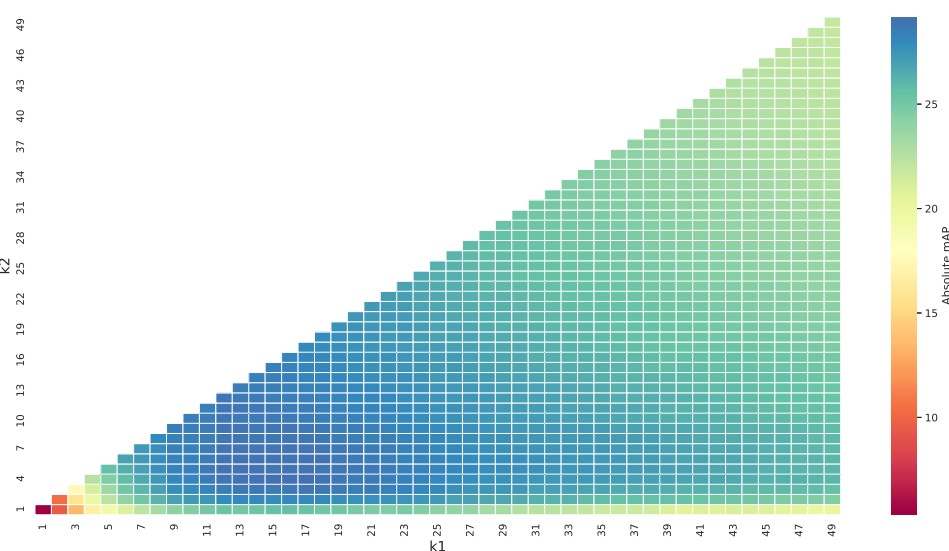

**Figure 9.** Heatmap of the absolute mAP against *k*1 and *k*2 with the MD method (best seen in color).

Similarly, Figure 10 shows the evolution of the absolute mAP against $\alpha$, with $(k1, k2)$ fixed to $(15, 4)$. Again, the performance does not significantly changes when $\alpha > 4$. This is a behavior similar to what was observed in the original proposition of $\alpha$QE [21], on which our proposed MD and cMD methods draw inspiration.

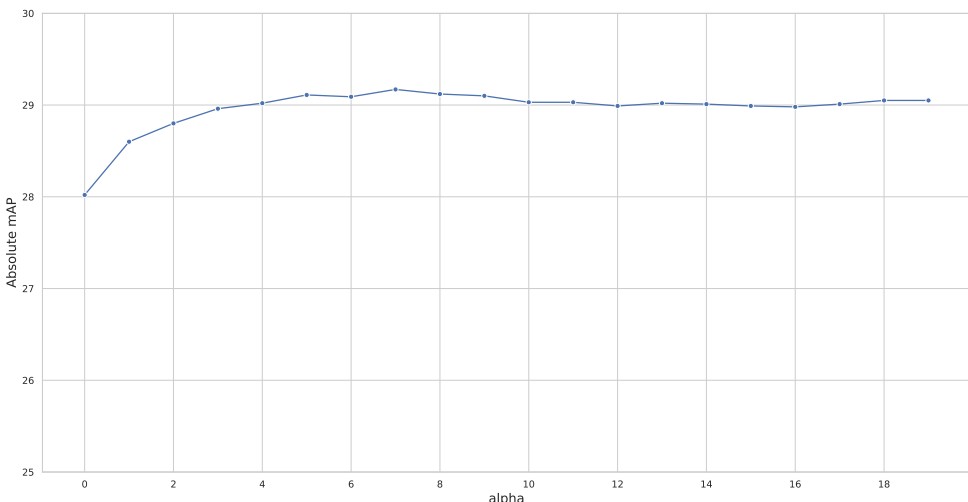

**Figure 10.** Evolution of the absolute mAP against $\alpha$ with the MD method.

### 4.4. Heterogeneity

We note that increased absolute performance, i.e., descriptor accuracy, is generally accompanied with an increase of the mAPD measure, indicating positive images from different collections being pushed to the end of the list. This is coherent with our assumption that descriptors are very dependent on their training statistics, meaning that they are accurate only on images with known semantics.

Diffusion does not prevent this and only reinforces dispersion of cross-domain images: the highest mAPD score corresponds to the highest absolute mAP with our proposed MD diffusion method.

To solve this challenge, we proposed the cMD method to study if it is possible to enhance cross-domain performance while keeping a reasonable absolute performance. In particular, the $\lambda$ parameter inserts control on the compromise between these two objectives. Figure 11 shows the influence of varying $\lambda$. We observe a bell-shaped curve for qP1 and

mP1, indicators of how high on the list positive cross-domain image are, while mAPD continuously decreases with increasing $\lambda$. It seems that there is a "sweet spot" for balancing absolute and cross-domain accuracy, around 0.5–0.6.

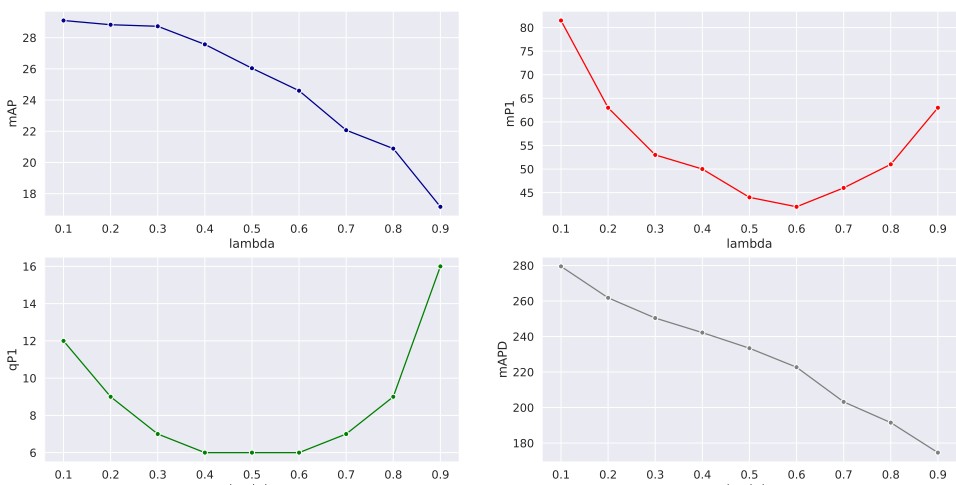

**Figure 11.** Evolution of the absolute mAP and various inter-domain measures against $\lambda$ with the cMD method.

Figure 12 shows some visual examples of the trade-off between accuracy and cross-domain retrieval: we observe that MD significantly improves the top 4 results with this query but mostly retrieves images in the same collection (here, internet). Our proposed cMD variation maintains the accuracy of the results but pushes positives from different collections higher on the list.

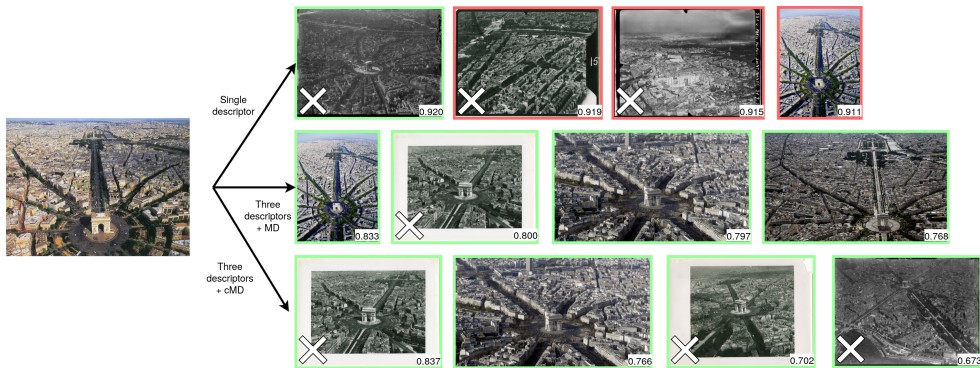

**Figure 12.** Query examples on the "Arc de Triomphe" class. First row: results using GeM-ArcFace trained on GoogleLandmarks (best performing single descriptor in Table 3). Second row: results using the three best performing descriptors + our proposed multi-descriptor diffusion (MD). Third row: results using the three best performing descriptors + our proposed constrained multi-descriptor diffusion (cMD). Positive results are indicated in green, negative in red. Similarity scores are indicated in the bottom right of each image. Cross-collection images (i.e., belonging to different collections) are indicated with a white cross on the bottom left.

### 4.5. Influence of the Training Dataset

The comparison between GeM trained on GoogleLandmarks and GeM trained on SF300 highlights the importance of the training dataset. The difference of 10 points of mAP is explained by the high volume and higher relevance of images in GoogleLandmarks. However, the method trained on SF300 is slightly better on Photothèque images, i.e., mostly vertical aerial imagery. This comparative difference justifies using multiple descriptors to handle multiple representation modalities.

### 4.6. Computational Complexity

Our proposed MD and cMD methods are agnostic of the descriptors used, and only require as inputs the corresponding similarity matrices. In this work, we introduce and evaluate the idea of connecting images regardless of their source, and consider that the descriptors are already computed and stored for evaluating the considered diffusion methods. In an online setup, issuing an unseen query would require to compute its descriptors and similarities with the whole base. This process can be optimized: a common approach compatible both with local and global descs is Product Quantization [68], where descriptors are separated and binarized in sub-vectors for efficient storing; and the diffusion process can be decoupled in offline and online steps [29] by pre-computing similarity matrices and diffusion steps. For the ALEGORIA benchmark, such scale-up methods are not necessary considering the relatively low volume of images.

Table 6 shows the computational overhead of the considered diffusion methods. Our proposed MD and cMD diffusion methods use the optimized GPU implementation of GQE [27], offering computation times lower than $\alpha$QE (CPU) even with two additional descriptors.

**Table 6.** Computational overhead of diffusion methods, on ALEGORIA. Experiments made with an Intel i7-8700K CPU (3.7GHz), 32Go of RAM and a single NVIDIA RTX2080 Ti (12Go VRAM).

| Method | Computation Time |
|---|---|
| $\alpha$QE | 147 ms |
| GQE | 57 ms |
| MD | 128 ms |
| cMD | 140 ms |

## 5. Conclusions

Using our proposed benchmark, its annotations, and our proposed evaluation protocol for cross-domain retrieval, we highlighted some important findings: reducing retrieval performance to a single global mAP values hides varying behaviors depending on the descriptor, and better absolute performance can degrade inter-domain performance.

Our experiments on diffusion follow the same observations, with increased performance but also increased specificity. Motivated by disparities in intra-domain performance and the modular setup of diffusion under the graph view, we proposed a new multi-descriptor diffusion method and a variation allowing the balance of absolute performance versus inter-domain performance. Results validate the effectiveness of our method.

Studying the case of ALEGORIA allows us to put the deep learning framework into perspective: the lack of relevant training datasets for specific tasks is not a fatality and can be mitigated with diffusion, and as suggested by our proof of concept with a sophisticated few-shot learning method, with meta-learning hopefully soon.

**Author Contributions:** Conceptualization, D.G., V.G.-B. and L.C.; methodology, D.G., V.G.-B. and L.C.; software, D.G.; validation, D.G.; formal analysis, D.G.; investigation, D.G.; resources, V.G.-B. and L.C.; data curation, D.G. and V.G.-B.; writing—original draft preparation, D.G.; writing—review and editing, D.G. and V.G.-B.; visualization, D.G.; supervision, V.G.-B. and L.C.; project administration, V.G.-B.; funding acquisition, V.G.-B. All authors have read and agreed to the published version of the manuscript.

**Funding:** This work is supported by ANR, the French National Research Agency, within the ALEGORIA project, under Grant ANR-17-CE38-0014-01.

**Data Availability Statement:** Data available on the project website: alegoria.ign.fr (accessed on 1 July 2021).

**Conflicts of Interest:** The authors declare no conflict of interest.

# Appendix A

**Table A1.** Details of the classes in the ALEGORIA benchmark.

| Class Name | Urban/Natural | Class Definition | Class Type |
|---|---|---|---|
| amiens | urban | tower | object |
| annecy | semi urban | lake mouth | zone |
| arc de triomphe | urban | monument | object |
| basilique sacre coeur | urban | church | object |
| biarritz | semi urban | hotel, beach | zone |
| amiral bruix boulevard | urban | crossroad | zone |
| bourg en bresse | semi urban | factory | object |
| brest | urban | port | zone |
| fourviere cathedral | urban | church | object |
| reims cathedral | urban | church | object |
| saint etienne de toul cathedral | urban | church | object |
| deauville international center | urban | hotel | object |
| charlevilles mezieres | urban | square | zone |
| chantilly castle | semi urban | castle | object |
| palace of versailles | urban | castle | object |
| choux creteil | urban | tower | object |
| cite internationale lyon | urban | neighborhood | zone |
| foix | semi urban | castle | object |
| gare du nord paris | urban | train station | object |
| gare est paris | urban | train station | object |
| gare perrache lyon | urban | train station | object |
| grenoble | urban | river | zone |
| guethary | natural | hotel | object |
| saint laurent hospital chalon | urban | hotel | object |
| nantes island | urban | neighborhood | zone |
| invalides | urban | hotel | object |
| issy moulineaux | urban | bridge | object |
| la madeleine paris | urban | monument | object |
| le havre | urban | tower | object |
| lery seyne sur mer | semi urban | church | object |
| macon | urban | bridge | object |
| mairie lille | urban | tower | object |
| chasseneuil memorial | natural | monument | object |
| mont blanc | natural | mountain | object |
| mont saint michel | natural | neighborhood | zone |
| neuilly sur seine | urban | neighborhood | zone |
| notre dame de lorette | natural | church | object |
| notre dame garde | urban | church | object |
| notre dame paris | urban | church | object |
| pantheon paris | urban | monument | object |
| picpus | urban | neighborhood | zone |
| place bourse bordeaux | square | square | zone |
| place marche clichy | urban | square | zone |
| bouc harbour | semi urban | harbor | zone |
| porte pantin | urban | neighborhood | zone |
| porte saint denis | urban | monument | object |
| aubepins neighborhood | urban | neighborhood | zone |
| reims racetrack | urban | neighborhood | zone |
| riom | urban | neihgborhood (town) | zone |
| saint claude | semi urban | church | object |
| gerland stadium | urban | monument | object |
| st tropez | semi urban | neighborhood (town) | zone |
| toulon | urban | neighborhood | zone |
| eiffel tower | urban | tower | object |
| tours | urban | neighborhood | zone |
| aillaud towers nanterre | urban | tower | object |
| vannes | urban | neighborhood | zone |
| villa monceau | urban | neighborhood | zone |

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
