# Peer review of "Connecting Images through Sources: Exploring Low-Data, Heterogeneous Instance Retrieval"

_remotesensing, doi:10.3390/rs13163080_

Round 1

Reviewer 1 Report

The manuscript proposed a review of cross-domain content-based image retrieval methods, where images could be from different dates, vertical or oblique aerial photographs, or modern street-level pictures. Authors centered on descriptors in these methods to exploit their strengths.

  • The paper should have a thorough proofreading to correct typos/spells. For example, litterature (lines 74, 129, 132, 321) -> literature; consists in (line 192) -> consists of; discuted (line 203) -> discussed
  • Annotated ALEGORIA dataset will be a great resource to the community. Lines 99-104 described the attributes used for annotation. The authors stated that they are quantized with integer values from 0 to 2 or 3. However, they seem very vague, especially scale, illumination, and level of occlusion. Further description with examples should be added.
  • Line 110: what are these 6 data sources?
  • Lines 146-152: What is the set S for R(i, S)? Why is there 1860 queries over the set Q? I assume the ranking of the image is based on the similarity of descriptors. But how they are calculated is not clear. Examples should be included in Section 4 during the discussion for Figure 8.
  • Lines 414-415: it is not clear in test setup E what noise is. Clarification should be added.
  • Overall, the paper seems interesting. However, for any new query image, descriptors are required to be extracted using the authors’ provided model weights. The computation complexity should be investigated in the paper. This aspect is critical to make the annotated ALEGORIA dataset benefit the research community.

Author Response

We thank you for your valuable suggestions and constructive feedback. Accordingly, we have made modifications on the manuscript to reflect these suggestions. In the following, we also respond point-by-point with references to the modifications (line numbers correspond to the new version, original remarks in bold). We hope that these explanations will clarify the methodological and technical choices we have made in this work.

    Annotated ALEGORIA dataset will be a great resource to the community. Lines 99-104 described the attributes used for annotation. The authors stated that they are quantized with integer values from 0 to 2 or 3. However, they seem very vague, especially scale, illumination, and level of occlusion. Further description with examples should be added.
    => We changed Figure 2 to give examples of attribute values, and the paragraph describing attributes (lines 134 to 142) to detail the possible values. We removed the mention of quantized integer values, it is how they are represented in the dataset .csv but they are not necessary for comprehension here.

    Line 110: what are these 6 data sources?
    => Our data comes from GLAMs (Galleries, Librarires, Archives, Museums), where images are encapsulated in collections. There are in total 5 collections + we add images from internet (6th collection). We added a description of the 5 collections in subsection 2.1.1 (lines 109-114), and clarified the notion of collection in the introduction (line 47). We also clarified the difference between institutions and collections in lines 105-114.

    Lines 146-152: What is the set S for R(i, S)? Why is there 1860 queries over the set Q? I assume the ranking of the image is based on the similarity of descriptors. But how they are calculated is not clear. Examples should be included in Section 4 during the discussion for Figure 8.
    => 1860 is a typo, it's actually 1858 queries. We clarified the notation in subsection 2.1.3 (lines 212-215). We added some clarification on how similarities are computed in subsection 2.5 (lines 459-461) but do not elaborate on this since it is descriptor-specific, and a new Figure 10 to give a visual example of retrieval with similarity scores included.

    Lines 414-415: it is not clear in test setup E what noise is. Clarification should be added.
    => We described how setup E uses noise (lines 484-486). Hopefully it is more clear.

        Overall, the paper seems interesting. However, for any new query image, descriptors are required to be extracted using the authors’ provided model weights. The computation complexity should be investigated in the paper. This aspect is critical to make the annotated ALEGORIA dataset benefit the research community.
    => For our un- and semi-supervised implementations of SimCLR and CNAPS, we do not think that the obtained results justify making model weights available (base architectures are downloadable here: SimCLR and CNAPS), and rather present them as exploratory works. For the supervised methods we compare, all methods with a star are available on GitHub and authors provide the weights (e.g. HOW , GeM-SfM120k). Our own implementation of GeM-ArcFace trained on GoogleLandmarks is not a contribution, and training scripts are easy to find for Pytorch or Tensorflow. Following our experiments on diffusion and the conclusion that using different descriptors with different characteristics might be a better approach than searching a single optimal descriptor, we prefer to let the readers interested in Alegoria experiment with their own approach (or use some of the many already available descriptors).
    => We added subsection 4.6 (lines 618-633) to discuss on the computational overhead of the compared diffusion methods and propose ways of scaling our approach to bigger datasets.

In addition to the above comments, we have corrected all English errors you have pointed out and thoroughly proof-read the paper. 
We hope that we have properly addressed and answered to all of your suggestions, and look forward to hearing from you or respond to any further questions or remarks you may have.

Sincerely,

Authors

Reviewer 2 Report

It is interesting to explore heterogeneous image retrieval. Besides, providing new benchmark dataset to the community can be of great benefit for everyone. However, the structure and content of this paper is slightly confusing, making it difficult for the reader to understand the importance and innovation of the paper. It is recommended that the introduction emphasizes the significance of the study (e.g., why there is a need to construct a multi-source cultural heritage dataset) and the current status of the study. Then, in a new Section, the basic information and characteristics of the new dataset, as well as the new requirements or challenges for retrieval are introduced in detail. In another Section, the corresponding retrieval methods, evaluation metrics, etc. are introduced according to the new requirements or challenges. Subsequently, the experimental process and results are described in detail. Other suggestions are shown below.

Where is Time in the title reflected in this paper? Please clarify.

Please state the importance and role of this new dataset in cultural heritage.

Please sort out the current status of datasets for image retrieval on cultural heritage or other fields.

What is the purpose of cross-searching between satellite images and photos (e.g., image from IGN and photo from Internet in Figure 2)? The features of these two types of images differ significantly in terms of cultural heritage, making it difficult to retrieve them by image content.

What are the 58 categories in the dataset? Please clarify in the form of an appendix.

As shown in Table 1, each category contains an average of only 31 images. However, the training of deep convolutional neural networks (CNNs) requires a large amount of sample data. Therefore, the new dataset clearly does not meet the requirements of CNNs. So how can others use the new dataset to train retrieval models?

Most of the experiments in Table 3 use other data sets as training sets. Why not use the new dataset as the training set to carry out the evaluations? Is it because of the small number of new dataset types described above? If so, how should the new datasets be used in practice?

Author Response

We thank you for your valuable suggestions and constructive feedback. Accordingly, we have made modifications on the manuscript to reflect some of these suggestions. In the following, we also respond point-by-point with references to the modifications (line numbers correspond to the new version, original remarks in bold). We hope that these explanations will clarify the methodological and technical choices we have made in this work.

    Where is Time in the title reflected in this paper? Please clarify.
    => It was indeed not sufficiently explored in the paper. We removed "Time" in the title to avoid the unnecessary emphasis, "Sources" being already a keyword covering the various acquisition conditions. Concurrently, we added a description of the collections used to build the dataset with their respective acquisition dates (lines 109-114).

    Please state the importance and role of this new dataset in cultural heritage.
    Please sort out the current status of datasets for image retrieval on cultural heritage or other fields.
    => We added two paragraphs in the introduction to better explain our motivations regarding cultural heritage (lines 17-39). We also added subsection 2.1.2 to position the Alegoria dataset in the context of cultural heritage datasets (lines 176-201). Note that we split our overview of datasets in two categories: first datasets with related motivations and semantics in section 2.1.2 (cultural heritage, long term image matching), and datasets that might serve as training datasets for building models with good performance on Alegoria in Table 2. 

    What is the purpose of cross-searching between satellite images and photos (e.g., image from IGN and photo from Internet in Figure 2)? The features of these two types of images differ significantly in terms of cultural heritage, making it difficult to retrieve them by image content.
    => Images from IGN are not satellite images, they are aerial photographs captured through various acquisition campaigns throughout the 20th century. We include them because they have better annotation standards, and are thus useful references to identify contents in other aerial photography collections which often do not include metadata. We describe this goal in the new subsection 2.1.2, and shortly in lines 115-117 when presenting the collections.

    What are the 58 categories in the dataset? Please clarify in the form of an appendix.
    => We added Table A7 in appendix A to show our class definitions. More statistics will be available on the project website (alegoria-project.fr/Alegoria_dataset).

    As shown in Table 1, each category contains an average of only 31 images. However, the training of deep convolutional neural networks (CNNs) requires a large amount of sample data. Therefore, the new dataset clearly does not meet the requirements of CNNs. So how can others use the new dataset to train retrieval models?
    Most of the experiments in Table 3 use other data sets as training sets. Why not use the new dataset as the training set to carry out the evaluations? Is it because of the small number of new dataset types described above? If so, how should the new datasets be used in practice?    
    => The Alegoria dataset is a benchmark and is indeed not suited for training CNNs. We added some words to clarify this in the introduction of section 2.3 (lines 282-283). We propose this benchmark to evaluate how modern state-of-the-art CNN-based descriptors behave on a topical area (cultural heritage and more specifically content describing the territory, see section 2.1.2), especially in cross-domain (or cross-collection) situations, and in real conditions of use (what to do with CNNs when training data are lacking). We also propose new measures related to the cross-domain retrieval problem. To solve the lack of training data, section 2.3 proposes some strategies with un- and semi-supervised training.

We hope that we have properly addressed and answered to all of your suggestions, and look forward to hearing from you or respond to any further questions or remarks you may have.

Sincerely,

Authors

Round 2

Reviewer 1 Report

The revised manuscript has addressed my comments and concerns. Recommend to accept it for publication.

Author Response

We thank you for your positive feedback.

Sincerely,

Authors

Reviewer 2 Report

The manuscript improved a lot. Some additional suggestions and comments are shown below.

Abstract should also be introduced from the perspective of cultural heritage.

The authors confirmed that the benchmark dataset Alegoria is indeed not suited for training CNNs. This limits the application of  the benchmark dataset Alegoria. Besides, although the authors proposed some measures to improve the accuracy, the value of mAP is still very low and cannot meet the requirements of practical applications. It is hoped that the author can expand the number of images in the Alegoria dataset in the future. It is also hoped that the authors can evaluate the performance of low and medium level features on the dataset.

Author Response

We thank you for your comments and thoughtful suggestions. We modified the abstract to better introduce the paper in the context of cultural heritage.

We would like to add a few comments on the additional remarks (in bold):

    The authors confirmed that the benchmark dataset Alegoria is indeed not suited for training CNNs. This limits the application of the benchmark dataset Alegoria.

    => An important starting point of our paper is that deep learning has currently both the advantage and the drawback of being extremely data-driven. This raises important questions about how to generalize to situations where training data is not available (including many other fields than cultural heritage, e.g. medical imagery, robotics, specific object detection & counting...). In our case, while we are aware that a straightforward solution would have been to invest money for annotating enough images and train a CNN, we believe it would not have been a significant research contribution because it would only have answered our specific case of territorial historical imagery. Rather, similarly to the recent trend of meta-learning, we think it is important to stimulate research to produce methods that are less data-dependent and can generalize to any situation, and we hope that the Alegoria benchmark and our proposed approach to the problem of low-data, heterogeneous retrieval will help to formulate and evaluate new methods.

    Besides, although the authors proposed some measures to improve the accuracy, the value of mAP is still very low and cannot meet the requirements of practical applications.

    => The mAP being, in absolute, "low" (far from its optimal value of 100%), is not in itself a problem. The benchmark is voluntarily made very challenging to foster new methods answering the problem of heterogeneous, low-data image retrieval. Note for example the maximum performance of 38.68% obtained on the 2020 Google Landmark Retrieval Kaggle Challenge. Similarly, it is precisely because the performance on Oxford and Paris was saturating (>90%) that the authors of the revisited version of these datasets proposed new annotations and evaluation setups, and undoubtedly stimulated new research, including the Google Landmark Retrieval Challenge.

    It is hoped that the author can expand the number of images in the Alegoria dataset in the future

    => The Alegoria benchmark presented in this paper, that we manually annotated to study the potential of content-based image retrieval, is only a small part of the data of a bigger research project also including works on metadata and 3D modelization (>80k images). In a second phase, we will enrich the Alegoria benchmark with these data.

    It is also hoped that the authors can evaluate the performance of low and medium level features on the dataset.

    => The paper does include the local descriptor HOW and the regional descriptor RMAC, by architecture design they are restricted to describing low or medium level visual patterns (edges, corners, any discriminative local pattern). If you mean hand-crafted descriptors, we compared the ORB descriptor (low level, similar to SIFT) in the preliminary paper on ALEGORIA and concluded that it cannot compete with modern deep learning-based features. This is mentioned in lines 383-388 of the paper.

Once more, we are deeply thankful for your careful reading, insightful comments and constructive suggestions, which have helped a lot to make the contributions of our manuscript stronger.

Sincerely,

Authors